# Cochayuyo (*Durvillaea incurvata*) Extracts: Their Impact on Starch Breakdown and Antioxidant Activity in Pasta during In Vitro Digestion

**DOI:** 10.3390/foods12183326

**Published:** 2023-09-05

**Authors:** Luz Verónica Pacheco, Javier Parada, José R. Pérez-Correa, María Salomé Mariotti-Celis, Mario Simirgiotis

**Affiliations:** 1Graduate School, Faculty of Agricultural and Food Sciences, Universidad Austral de Chile, Valdivia 5090000, Chile; luz_pacheco007@hotmail.com; 2Institute of Food Science and Technology, Faculty of Agricultural and Food Sciences, Universidad Austral de Chile, Valdivia 5090000, Chile; 3Department of Chemical and Bioprocess Engineering, Pontificia Universidad Católica de Chile, Santiago 7810000, Chile; perez@ing.puc.cl; 4Escuela de Nutrición y Dietética, Universidad Finis Terrae, Santiago 7501015, Chile; mmariotti@uft.cl; 5Instituto de Farmacia, Facultad de Ciencias, Universidad Austral de Chile, Valdivia 5090000, Chile; mario.simirgiotis@uach.cl

**Keywords:** phlorotannin, starch digestion, edible seaweed, anti-diabetic effect

## Abstract

Seaweeds, notably cochayuyo (*Durvillaea incurvata*), are recognized for their rich macro- and micronutrient content, along with their inhibitory effects on the α-glucosidase enzyme. The present study aims to evaluate the effectiveness of this inhibition in actual starchy food products under in vitro gastrointestinal conditions. This study utilized freeze-dried cochayuyo, extracted using hot pressurized liquid extraction with 50% ethanol at 120 °C and 1500 psi. The inhibition mechanism of α-glucosidase was determined, and the polyphenol composition of the extract was analyzed using Ultra-High-Performance Liquid Chromatography. This study further evaluated the extract’s impact on starch digestibility, total phenolic content, and antioxidant capacity in pasta (noodles) as representative starchy food under gastrointestinal conditions. The results indicate that the α-glucosidase inhibition mechanism is of mixed type. Phenolic compounds, primarily tetraphloroethol, could contribute to this anti-enzymatic activity. The extract was observed to decrease starch digestibility, indicated by a lower rate constant (0.0158 vs. 0.0261 min^−1^) and digested starch at an infinite time (77.4 vs. 80.5 g/100 g). A significant increase (~1200 vs. ~390 µmol TROLOX/100 g) in antioxidant activity was also noted during digestion when the extract was used. Thus, this study suggests that the cochayuyo extract can reduce starch digestion and enhance antioxidant capacity under gastrointestinal conditions.

## 1. Introduction

The seaweed *Durvillaea incurvata*, endemic macroalgae from Chile, is commonly known as “cochayuyo” or “coyofe” among the Chilean population. It has been used as food for several thousand years, as suggested by archeological findings conducted in southern Chile [1]. Marine algae are generally considered a nutritious and healthful food source because they contain macro- and micronutrients (proteins, peptides, amino acids, polysaccharides, phenolic compounds, lipids, vitamins, and minerals) [2].

Brown seaweeds (Phaeophyceae) like *Durvillaea incurvata* have garnered significant interest due to their phlorotannins (phloroglucinol units-based molecules) content. Phlorotannins are phenolic compounds associated with various health benefits, including antiviral, anti-diabetic, antimicrobial, anti-inflammatory, anticancer, antioxidant, and anti-allergic properties [3].

Numerous studies have demonstrated that phlorotannins can reduce the glycemic impact of starchy foods, making them beneficial for individuals with diabetes. This effect is achieved by inhibiting the enzymes α-amylase and, mainly, α-glucosidase, which slow down carbohydrate digestion [3,4]. Additionally, phlorotannins can enhance postprandial glucose clearance by increasing insulin-mediated glucose uptake in skeletal muscle by activating the PI3K/PKB pathway, helping with glycemic homeostasis [5]. Therefore, phlorotannins and other components found in seaweed can be considered attractive ingredients for developing healthier foods. They can potentially contribute to combating hyperglycemia-related diseases, such as type 2 diabetes. The worldwide prevalence of type 2 diabetes was ~9.3% (~463 million people) in 2019, and it is estimated to increase to ~10.2% (~578 million) by 2030 and ~10.9% (~700 million) by 2045 [6].

When designing foods with beneficial health properties, it is important to highlight the potential of pasta as a food matrix for incorporating bioactive compounds. In general, it has been observed that pasta can be fortified with supplements from various sources rich in bioactive compounds, thereby enhancing its nutritional properties [7].

Previous research of our group showed that *Durvillaea incurvata* extract obtained by hot pressurized liquid extraction (ethanolic solution as solvent) could strongly inhibit α-glucosidase (IC_50_ = 473.4 µg/mL), even more than the pharmacological inhibitor acarbose (IC_50_ = 797.85 µg/mL), although real impact on starchy food, as well as the details regarding the extract and type of inhibition, were not studied [8].

The main aim of the present research was to determine whether the observed inhibitory capacity of *Durvillaea incurvata* extract obtained by hot pressurized liquid extraction with an aqueous ethanol mixture applies to a starchy food system, thereby slowing down starch digestibility. In addition, to better understand this phenomenon, the type of α-glucosidase inhibition, the phenolic compound profile of the extract, and the changes in antioxidant capacity during digestion were also studied.

## 2. Materials and Methods

Unless otherwise stated, all chemicals and cell culture reagents were obtained from Sigma Chemical Co. (Saint Louis, MO, USA).

### 2.1. Seaweed Collection

Samples of seaweed *Durvillaea incurvata* (cochayuyo) were collected in September 2017 from the “Palo Muerto” location, commune of Corral, at “Region de Los Rios”, Southern Chile. Samples were quickly washed in cold water to remove sand and other particles, cut to ~1 cm^3^, and immediately frozen and stored at −80 °C until freeze drying.

### 2.2. Phenolic Compounds Extraction

Phenolic compounds were extracted using two different methods: hot pressurized liquid extraction and atmospheric solid–liquid extraction. Aqueous ethanol (50% *v*/*v*; eco-friendly method) and aqueous acetone (60% *v*/*v*; as control) mixtures were used as solvents for each respective extraction method. For both methods, freeze-dried seaweed material was mixed with the extraction solvent at 1:32 *w*/*v*. The hot pressurized liquid extraction using an aqueous ethanol mixture (ethanol extract) was performed in an Accelerated Solvent Extractor (Thermo Scientific™ Dionex™ ASE™ 150) at 120 °C and 1500 psi. After extraction, ethanol extracts were transferred to an amber pet bottle and stored at −20 °C. Seaweed samples were extracted using an aqueous acetone mixture for 1 h on a thermoregulatory rotary shaker at 100 rpm and 30 °C for the atmospheric solid–liquid extraction. Then, the mixture was centrifuged for 5 min at 6000 rpm and 20 °C, and the supernatant was transferred to a 50 mL flask wrapped in aluminum. Water was added to the remaining solid, stirred manually, and centrifuged. The supernatant was transferred to the previous flask and adjusted to 50 mL with 60% acetone. Finally, the extract was transferred to an amber pet bottle and stored at −0 °C. Before all analysis, each extract was dried with an air pump until the solvent evaporated.

### 2.3. Type of Inhibition for α-Glucosidase by Extracts

Stock solutions were prepared as follows: Each dry extract was dissolved in DMSO overnight, where the final concentration of the solvent in the stock solution was 10 mg/mL. Equivalent quantities of DMSO were also added to control samples. Each extract was then filtered using a 0.45 μm syringe filter. Each stock solution was diluted with phosphate buffer 100 mM pH 6.9 to give the desired working concentration. For each independent experiment, all samples were tested in triplicate.

Extracts were held at a constant concentration (1000 μg/mL phosphate buffer 100 mM pH 6.9) and incubated in the presence of increasing concentrations of p-nitrophenyl-α-D-glucopyranoside (PNPG) (1, 2, 3, 5, 7 10 mM). The samples’ inhibition modes against α-glucosidase were determined according to the method described by Lordan et al. [9]. Briefly, 50 μL of extract solution and 50 μL of each PNPG solution were mixed in a 96-well microplate and incubated at 37 °C for 5 min. Then, phosphate buffer (100 μL) containing 0.1 U/mL of α-glucosidase (from *Saccharomyces cerevisiae*) was added to each well. Absorbance at 405 nm was recorded using a microplate reader at 37 °C for 15 min. Finally, the data were analyzed by the Lineweaver–Burk diagram.

### 2.4. Polyphenols Profile in the Ethanol Extract by UHPLC

Ultra-High-Performance Liquid Chromatography (UHPLA) was applied to identify the polyphenols in the sample. A Thermo Scientific Dionex Ultimate 3000 UHPLC system, hyphenated with a Thermo high-resolution Q-Exactive focus mass spectrometer (Thermo, Bremen, Germany), was used for the analysis. The chromatographic system was coupled to the MS with a Heated Electrospray Ionization Source II (HESI II). Nitrogen (purity > 99.999%) obtained from a Genius NM32LA nitrogen generator (Peak Scientific, Billerica, MA, USA) was employed to produce MS fragmentation. Mass calibration for Orbitrap was performed once a day, in both negative and positive modes, to ensure a working mass accuracy lower than 5 ppm. Ultramark 1621 (Alpha Aezar, Stevensville, MI, USA) mixed with caffeine and N-butylamine (Sigma Aldrich, Saint Louis, MO, USA) plus buspirone hydrochloride, sodium dodecyl sulfate, and taurocholic acid sodium salt (Sigma Aldrich, Saint Louis, MO, USA), was used as a standard solution for calibration. These compounds were dissolved in a mixture of acetic acid, acetonitrile, water, and methanol (Merck, Darmstadt, Germany) and were infused using a Chemyx Fusion 100 syringe pump (Thermo Fisher Scientific, Bremen, Germany). XCalibur 3.0 software (Thermo Fisher Scientific, Bremen, Germany) and Trace Finder 3.2 (Thermo Fisher Scientific, San José, CA, USA) were used for UHPLC control and data processing, respectively. Q Exactive 2.0 SP 2 from Thermo Fisher Scientific was used to control the mass spectrometer.

Solvent delivery was performed at a flow rate of 1 mL/min. Ultra-pure water with 1% formic acid (A) and acetonitrile with 1% formic acid (B) were used. The elution started with 5% B at zero time. After that, the system maintained 5% B for 5 min, then transitioned to 30% B within 10 min. The system then maintained 30% B for 15 min before moving to 70% B for 5 min. Afterward, the system maintained 70% B for 10 min. Finally, the system returned to 5% B for 10 min and maintained this condition for an additional 12 min to stabilize the column (HPLC C18; Thermo Fisher Scientific, Waltham, MA, USA) before the next injection of 20 µL.

The HESI parameters were as follows: sheath gas flow rate, 75 units; auxiliary gas unit flow rate, 20; capillary temperature, 400 °C; auxiliary gas heater temperature, 500 °C; spray voltage, 2500 V (for ESI-); and S lens, RF level 30. Full scan data in positive and negative modes were acquired at a resolving power of 70,000 FWHM (full-width half maximum) at *m*/*z* 200. A scan range of *m*/*z* 100–1000 for the compounds of interest was chosen; the automatic gain control (AGC) was set at 3 × 106, and the injection time was set to 200 ms. The scan rate was set at 2 scans s^−1^. External calibration was performed using a calibration solution in positive and negative modes. For confirmation purposes, a targeted MS-MS analysis was performed using the mass inclusion list, with a 30 s window, with the Orbitrap spectrometer operating in positive and negative modes at 17,500 FWHM (*m*/*z* 200). The AGC target was set to 2 × 105, with a maximum injection time of 20 ms. The quadrupole filtered the precursor ions, which operated at an isolation window of *m*/*z* 2. The fore, high, and ultrahigh vacuum were maintained at approximately 2 mbar, from 105 and below 1010 mbar, respectively. Collision energy (HCD cell) was operated at 30 kV. Detection was based on the calculated exact mass and on the retention time of compounds. The mass tolerance window was set to 5 ppm for the two modes.

### 2.5. Impact of Phenolic Extracts on Starchy Food Digestion

To investigate the inhibitory effect of phenolic compounds on starch digestion of foods, wheat pasta (noodles purchased from a local supermarket) was chosen as the model food. The declared composition of the pasta included energy content (338 kcal/100 g), protein (11.0 g/100 g), fat (2.0 g/100 g), available carbohydrates (69.0 g/100 g), total sugar (4.0 g/100 g), and total fiber (3.9 g/100 g). The digestion of starch and changes in the total phenolic content and antioxidant capacity in digesta fluids were studied throughout the digestion process.

#### 2.5.1. Starch Digestion

Extract solutions were prepared as follows: 20 mg of each dry extract was dissolved in 20 mL of 100 mM sodium phosphate buffer (pH 6.9) just before assay. The pasta was cooked according to product directions. After that, 5 g of cooked pasta at room temperature (18 °C) was added to each extract solution (1 mg/mL buffer phosphate 100 mM pH 6.9). For digestions, three different conditions were considered: (i) Cooked pasta in 20 mL of phosphate buffer (without extract), (ii) Cooked pasta + ethanol extract solution, (iii) Cooked pasta + acetone extract solution (control extract).

The gastrointestinal system was based on the procedure described by Bellesia et al. [6], with slight modifications. There were three main phases distinguished: oral, gastric, and intestinal. The experiment lasted a total of 5.2 h, with durations of 0.2 h, 2.0 h, and 3.0 h for the oral, gastric, and intestinal stages, respectively. First, each substrate was homogenized in a laboratory mixer (Stomacher) for 1 min to simulate chewing in the presence of 5 mL of simulated salivary fluid. The artificial saliva consisted of a 0.1 M phosphate buffer (pH 6.9) containing 1.336 mmol/L of CaCl_2_, 0.174 mmol/L of MgSO_4_, 12.8 mmol/L of KH_2_PO_4_, 23.8 mmol/L of NaHCO_3_, and 150 units/L of α-amylase. After 10 min of incubation at 37 °C, 100 rpm, the pH was adjusted to 2.5 (to simulate the gastric pH) with concentrated HCl, and then 2.0 g/L of NaCl and 315 U/mL of pepsin were added. The solution was incubated at 37 °C in an incubator at 100 rpm for 2.0 h. At the end of the gastric digestion, the pH was brought to 7.5 with NaHCO_3_ (to simulate the hepato-pancreatic pH) before adding 0.8 g/L of pancreatin, 5 mg/mL of bile salts, and 2 mL of solution of α-glucosidase (10 U). Based on the added pancreatin, the amount of digestive enzymes in the intestinal fluid was 80 U/mL of α-amylase, 240 U/mL of proteases, and 384 U/mL of lipase. The solution was subsequently incubated at 37 °C in a shaking bath at 100 rpm for a further 3.0 h. The amount of glucose released was measured with an Accu-Check^®^ Performa^®^ glucometer (Roche Diagnostics SL, Barcelona, España). The glucose concentration in the digestion was measured within the range of the glucometer (0.6–33 mM L^−1^) at times 10, 20, 30, 45, 60, 90, 120, 150, and 180 min. The digested starch (*DS*) (g per 100 g of dry starch) at a measurement time (min) was calculated as follows:(1)$$DS=\frac{\left(0.9\times GR\times 180\times V\right)}{\left(W\times S\times \left[100-M\right]\right)}$$
where: 0.9 = stoichiometric constant for starch from the glucose content, *GR* = glucometer reading (mM/L), *V* = digestion volume (mL), 180 = glucose molecular weight, *W* = sample weight (g), *S* = starch content of the sample (g per 100 g of solids measured using the Starch test kit, catalog number SA20, Sigma-Aldrich, St. Louis, MO, USA), *M* = moisture content of the sample (g per 100 g).

The following model, adapted from Goñi et al. [10], was fitted to the outcomes, and *R*^2^ was obtained:(2)$$DS=D_{0}+D_{\infty -0}\left(1-e^{-kt}\right)$$
$$D_{\infty -0}=D_{\infty }-D_{0}$$
where *D*_0_ is the initial digested starch, *D_∞_* the digested starch at infinite time, *k* is the rate constant (1/min), and *t* is the digestion time (min).

#### 2.5.2. Changes in Total Phenolic Content and Antioxidant Capacity during Food Digestion

For analysis, fluid samples were collected at the previously described stages of starch digestion (start, oral, gastric, and intestinal). Each sample was centrifuged at 12,000 rpm at 24 °C for 10 min, and the supernatant was immediately analyzed. Outcomes are the “bioaccessible” phenolics/antioxidant capacity.

Total phenolic content

The total polyphenol content was determined using the Folin–Ciocalteu (FC) method with gallic acid as standard. In brief, 0.5 mL of sample or solvent blank was diluted in 3.75 mL of distilled water. Subsequently, 0.25 mL of FC reagent was added and homogenized. Then, 0.5 mL of sodium carbonate solution (10% *w*/*v*) was added and homogenized to react for 1.0 h at room temperature (18 °C). The absorbance of the reaction product was measured at 765 nm (UV spectrophotometer 1240, Shimadzu, Kyoto, Japan). The total polyphenol content was calculated as mg of gallic acid equivalents (mg GAE) per gram of dry seaweed, using a standard curve of 0.01–0.1 mg GAE/mL. Each extract was analyzed in duplicate.

Antioxidant capacity

The change in antioxidant capacity was analyzed using the free radical scavenging by Oxygen Radical Absorbance Capacity (ORAC) assay based on the procedure described by Cao and Prio [11], with minor modifications. The reaction was carried out in 75 mM phosphate buffer (pH 7.4) in a 96-well microplate. A 45 μL quantity of sample and 175 μL of fluorescein were deposited at 108 μL, and this mixture was incubated for 30 min at 37 °C; after that time, 50 μL of the AAPH solution was added to 108 μL. The microplate was immediately placed in the dual-scan microplate spectrofluorometer (Gemini XPS) for 60 min, and fluorescence readings were recorded every 3.0 min. The microplate was automatically shaken before and after each reading. For the calibration curve, Trolox was used at 6, 12, 18, and 24 μM. All reactions were carried out in triplicate. The area under the curve (AUC) was calculated for each sample by integrating the relative fluorescence curve. The net AUC of the sample was calculated by subtracting the AUC of the blank. The regression equation between the net AUC and the Trolox concentration was determined, and ORAC values were expressed as µmol Trolox equivalents (µmol TE) per gram of dry seaweed using the standard curve established previously.

### 2.6. Statistics

Data were analyzed by analysis of variance (ANOVA) followed by Tukey’s Multiple Comparison test. Software STATGRAPHICS Centurion XV.II (Old Tavern Rd, The Plains, VA, USA) was used.

## 3. Results and Discussion

Previously, our group demonstrated that an extract obtained using ethanol (using the same technique as in the present research and from the same seaweed) did not affect α-amylase activity. However, it exhibited a significant inhibitory effect on α-glucosidase, reaching nearly complete inhibition at 1000 µg/mL, with no observed toxicity [8]. Several analyses were performed to understand better this inhibition.

### 3.1. Type of α-Glucosidase Inhibition

The type of inhibition was determined by double reciprocal Lineweaver–Burk (Figure 1) graphics and according to the Km and Vmax parameters (Table 1). The extracts decreased the maximum velocity values (Vmax) and increased the constant Km compared to the control (free of inhibitor). It can be inferred that the inhibition of α-glucosidase observed is of the mixed type since adding the extracts results in changes in Km and Vmax. The inhibitor can bind to the enzyme, regardless of whether or not the enzyme has already bound to the substrate. Additionally, this inhibitor may bind to an allosteric site, which refers to a site other than the active site where the substrate is located. However, it is important to note that not all inhibitors binding to allosteric sites are considered mixed-type inhibitors. Mixed inhibition can lead to a decrease in the apparent affinity of the enzyme for the substrate (Km) regardless of whether the inhibitor prefers the free enzyme or enzyme–substrate complex. In any case, the inhibition reduces the apparent maximum enzymatic reaction rate [12].

In addition, there was a large decrease in the maximum reaction rate with both extracts (Table 1). Regarding Km, it could be inferred that the acetone extract is a stronger inhibitor than the ethanolic extract since, according to [13], an inhibitor is stronger when it shows a higher Km.

Our results coincide with previous studies where food-origin polyphenols have been described as mixed-type inhibitors, reducing the catalysis speed and accessibility to the active site of the substrate [7,14], although the actual degree of inhibition depends on the specific enzymes considered in the experimental model [12,15].

### 3.2. Profile of Polyphenols in the Ethanol Extract

As already published, several compounds were identified in the *Durvillaea incurvata* ethanol extract [16]. Figure 2 shows the chromatogram of the ethanol extract, while Table 2 shows the key data of the peaks. The detected compounds were sorbitol, 2,6,10,14-tetraoxapentadecane-4,12-diol (C_28_H_45_O_11_^−^), leucodelphinidin (C_15_H_13_O_8_^−^), stigmatellin (C_29_H_39_O_6_^−^), triplhoroethol (C_18_H_13_O_9_^−^), isomers of hexaphloroethol or bis-trifucophloroethol (C_36_H_25_O_8_^−^), tetraphloroethol (497.07364, C_24_H_17_O_12_^−^), pentaphloroethol or trifucophloroethol (C_30_H_21_O_15_^−^), and some which remain unidentified. These results confirm the presence of polyphenols, which explains both the antioxidant capacity and the inhibition of the enzymatic activity of the extracts. The most abundant compound was tetraphloroethol, while the presence of other compounds like sorbitol, along with the common presence of heavy metals, suggests that a purification step could be included if the achievement of a purified extract is mandatory [4].

### 3.3. In Vitro Digestion of Pasta Affected by Phenolic Compounds

It is important to highlight the possible differences in enzyme activity tests due to using a starch or synthetic substrate solution instead of real food. Using real foods implies the presence of a tridimensional food matrix where several types of molecules (such as proteins, lipids, and fibers), in addition to starch, can interact and diminish the effect of bioactive compounds on enzymes [17,18].

Noodles or pasta are an important source of carbohydrates, especially starch. A 100 g serving of raw pasta contains approximately 68 g of starch [19]. Therefore, starch is the most abundant nutrient in pasta and the quick/slow-release glucose source during digestion. Among carbohydrate-rich foods, noodles have a lower GI than bread, pizza, and other cereals [20].

The pasta was cooked in the same way that a normal consumer can make it at home. Pasta digestion occurred in the presence of each extract (ethanol and acetone) solubilized in a phosphate buffer pH 6.9. The effect of the extracts on starch digestion is shown in Figure 3. Starch digestion was not observed during the oral and gastric stages. Furthermore, in the absence of extracts and the presence of the ethanolic extract, starch digestion remained negligible until after the first 20 min of the intestinal phase. However, starch digestion was completely absent when the acetonic extract was utilized, persisting even after 30 min. As shown in the digestogram, clear differences in the starch digestion curves were observed between the absence and presence of the extracts. Glucose in the presence of acetone extract was detectable after 45 min of the intestinal phase. This delay in the hydrolysis of glycosidic bonds by the α-glucosidase for glucose release could be explained by the particular ability of the acetone extract to also inhibit α-amylase. This inhibition reduces the availability of oligosaccharides and disaccharides for α-glucosidase activity, consequently delaying the release of glucose to sufficient detection levels [8].

Regression analysis showed a good fitting of the model (Equation (1)) since *R*^2^ for the three data series was higher than 0.9 (see Table 3). The ethanol and acetone extracts decreased the total digestibility of starch, as shown by values of *D_∞_* (Table 3). The ethanol extract exhibited the slowest starch digestion rate, as indicated by its lower k value. Hence, the glucose release of the ethanol extract was comparatively slower than the acetone extract (Figure 3 and Table 3). Therefore, although both extracts make starch less digestible, the ethanol extract could be considered better for reducing the glycemic and insulinemic responses in vivo, since additionally to a lower total starch digestibility; such digestion is slower, that being crucial to achieving better homeostasis [21]. The differences between both digestograms are small (see Figure 3), suggesting small in vivo differences, especially considering the many additional factors determining the postprandial glycemic response. Eelderink et al. [22] concluded that slower intestinal uptake of glucose from a starchy food product could result in lower postprandial insulin and glucose-dependent insulinotropic polypeptide (GIP) concentrations but not necessarily in a lower glycemic response because of a slower glucose clearance rate (GCR).

The decrease in starch digestion due to the extracts can be attributed to phlorotannins, a predominant polyphenol group found in brown seaweeds such as *Durvillaea incurvata*. Phlorotannins have demonstrated the ability to inhibit the two enzymes involved in the intestinal digestion of starch, namely α-glucosidase and α-amylase. As a result, they can potentially reduce postprandial blood glucose levels and insulin spikes. Furthermore, these compounds have also been reported to exhibit other biological effects that influence glycemic homeostasis. For instance, they can enhance glucose uptake by skeletal muscles and inhibit protein tyrosine phosphatase 1B (PTP1B), a leptin and insulin signaling pathway regulator. Consequently, phlorotannins have the potential to improve insulin sensitivity [23]. Previous studies have shown the potential of seaweed to moderate blood glucose levels, associated with the capacity of several compounds to inhibit starch digestion enzymes, as well as various other mechanisms [24,25].

On the other hand, it has been shown that different brown seaweed extracts with varying composition and molecular weight distribution differentially inhibit α-glucosidase activities (enzyme kinetics and the mechanism of inhibition of maltase and sucrase), which may explain why different seaweed extracts generate an unequal inhibition of carbohydrate digestive enzymes [26].

Additionally, it should be considered that starch digestibility in foods is determined by several other factors such as starch source, amylose/amylopectin ratio, starch, and protein complexes, degree of gelatinization/retrogradation of starch, and presence of dietary fiber [27].

### 3.4. Bioaccessibility of Antioxidant Compounds from Durvillaea Antarctica Extracts during In Vitro Paste Digestion

Bioaccessibility is the amount of a compound released from its matrix in the gastrointestinal tract and available for absorption (for example, entering the bloodstream) [28]. The bioaccessibility of phenolic compounds during in vitro paste digestion was determined by measuring TP and antioxidant activity into digesta fluid at the end of each stage of digestion (oral, gastric, and intestinal). The results of both TP and antioxidant activity are shown in Figure 4.

TP increased through the digestion process for all samples, with or without seaweed extract, although in samples where the extract was present, the TP at the end of digestion was higher than in pasta alone. Outcomes suggest that through digestion, phenolic compounds are released and modified (e.g., fragmentation of high molecular weight phlorotannins to smaller ones) [29] and that pasta also contains such compounds. It should be noted that the breakdown of complex polyphenolic structures into smaller units has also been found to relate to the action of gut microbiota enzymes [30].

As an example of changes in the phenolic compounds during digestion, Huang et al. [31] studied the effects of in vitro simulated digestion on the free and bound phenolic content of seven seaweed species and showed that intestinal conditions could significantly increase the free phenolic content of all tested seaweeds; the total phenolic content of the seaweeds increased from 4.16 to 17.24 mg GAE/g before simulated digestion to 4.08 to 40.37 mg GAE/g after digestion, which is in line with our results.

It is known that changes in digestion conditions, such as pH or the presence of certain molecules, can affect phenolic compounds’ stability, and therefore they are somewhat unstable under real or simulated gastrointestinal conditions. For example, anthocyanins have been shown to degrade in pancreatic media [32], while ellagitannins may suffer partial degradation in the gastrointestinal tract [17].

Regarding the changes in ORAC antioxidant capacity throughout digestion (Figure 4B), the results showed a considerable difference between pasta with and without seaweed extracts. A mild increase was observed during the digestion of samples without extract, probably due to the pasta’s components, as suggested by TP outcomes (see Figure 4A). In contrast with this moderate increase, for samples having both seaweed extracts, the antioxidant capacity of the digesta was highly increased, reaching values several times higher than the initial one or the digested pasta alone. Such increased antioxidant capacity could be related to different modifications that the set of compounds within the algae extracts have suffered during digestion. Catarino et al. [33] suggested that phlorotannins may undergo different modifications during their transit in the gastrointestinal tract by enzymes like glucosidase enzymes, phase I enzymes, including cytochrome P450, and phase II enzymes (glucuronosyltransferases, sulfotransferases), and that resultant metabolites might represent active forms. Therefore, although the digestive process may affect the integrity and concentration of phlorotannins, this does not necessarily translate into the loss of bioactivity, probably owing to the bioactive effects that the degradation products of these phenolics may have [29].

This increase could also be associated with interactions of antioxidant compounds with food macromolecules (matrix effect). Specific polyphenols or other antioxidant compounds may have been released from molecules such as starch, other carbohydrates, or proteins through enzymatic action and pH changes. For example, non-covalent interactions (hydrogen bonds, hydrophobic interaction, and electrostatic and ionic interactions) between paste starch and antioxidant compounds such as polyphenols are possible [34]. Also, some antioxidants may be associated with alginic acid, a component of the cell wall of algae, through covalent bonds [35]. In any case, the increase in antioxidant capacity is a good indicator of the stability of the seaweed components and, therefore, its potential applications as a food ingredient. In general, the bioaccessibility of polyphenols depends on their release from the food matrix during digestion, changes in particle size, their hydrophilic/lipophilic balance as related to their glycosylation, different pH-dependent transformations (degradation, epimerization, hydrolysis, and oxidation within the gastrointestinal tract), and their interactions with food components. Additionally, there has been observed that the reduction in antioxidant capacity under intestinal conditions can be attributed to the structural reorganization of some compounds due to their sensitivity to alkaline pH [36].

Starch could interact with polyphenols, depending on the starch structure. In this sense, Gisbert et al. [37] found that polyphenols retention notoriously increased when starch gelatinization was carried out in the presence of seaweed flour. Polyphenols were physically adsorbed on the surface of the starch gel and, additionally, they were trapped inside starch gel walls, which in turn would be related to the liquid phase’s lower antioxidant capacity. This fact would explain, at least in part, the increase in antioxidant capacity during digestion as an effect of starch breakdown and the consequent release of polyphenols.

## 4. Conclusions

Our findings suggest that cochayuyo (*Durvillaea incurvata*) holds promise as a potential anti-diabetic food ingredient due to the inhibitory effects of its phenolic compounds on enzymes involved in starch breakdown, particularly α-glucosidase. The cochayuyo extract, obtained through hot pressurized liquid extraction (utilizing 50% ethanol at 120 °C and 1500 psi), effectively diminished the in vitro digestibility of starch in starchy foods. Notably, we observed an increased extract’s antioxidant activity under intestinal conditions, possibly resulting from molecular transformations in phlorotannins during digestion. Additionally, the α-glucosidase inhibition by the extract was determined to be of a mixed type. While these results are encouraging, further research, including in vivo studies on anti-diabetic effects, is necessary to design functional ingredients employing cochayuyo extracts. Nonetheless, the demonstrated effects on starch digestion and the bioaccessibility of its components underline the significant potential of cochayuyo in the realm of nutritional science.

## Figures and Tables

**Figure 1 foods-12-03326-f001:**
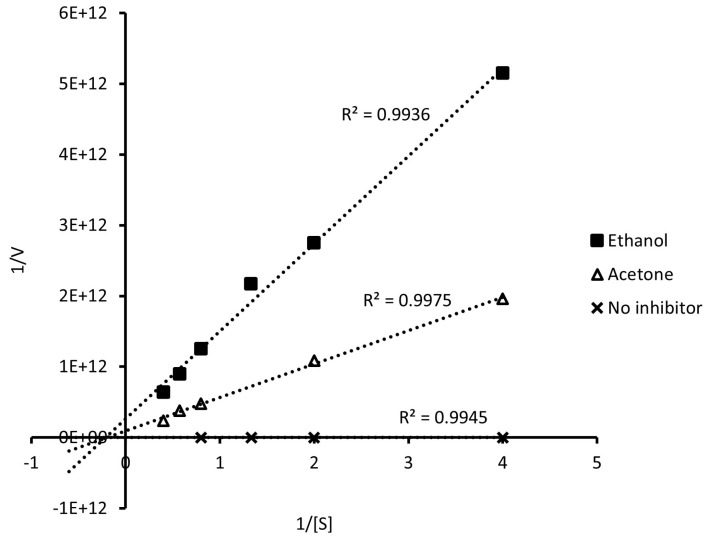
Lineweaver–Burk graphs: kinetics of α-glucosidase inhibition by ethanol–water and acetone extracts of *D. incurvata*.

**Figure 2 foods-12-03326-f002:**
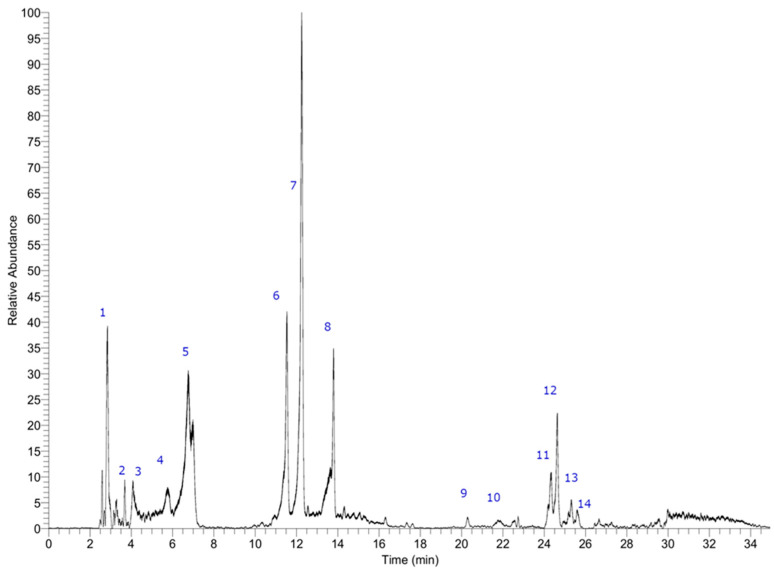
Chromatogram showing identified compounds in ethanol extract. Peak 1 was identified as sorbitol, peak 2 as 2,6,10,14-tetraoxapentadecane-4,12-diol (C_28_H_45_O_11_^−^), peak 3 as leucodelphinidin (C_15_H_13_O_8_^−^), peak 12 as stigmatellin (C_29_H_39_O_6_^−^). Peak 4 with an M-H- ion at *m*/*z*: 373.05685 was identified as triplhoroethol (C_18_H_13_O_9_^−^), peaks 6, 9, and 10 with ions at *m*/*z*: 745.10657, 745.10693, and 745.10675 as isomers of hexaphloroethol or bis-trifucophloroethol (C_36_H_25_O_8_^−^), peak 7 as tetraphloroethol (497.07364, C24H17O12^−^), and peaks 11 and 13 with parent ions at around 621 uma as pentaphloroethol or trifucophloroethol (C_30_H_21_O_15_^−^). Peaks 5, 8, and 14 remain unidentified.

**Figure 3 foods-12-03326-f003:**
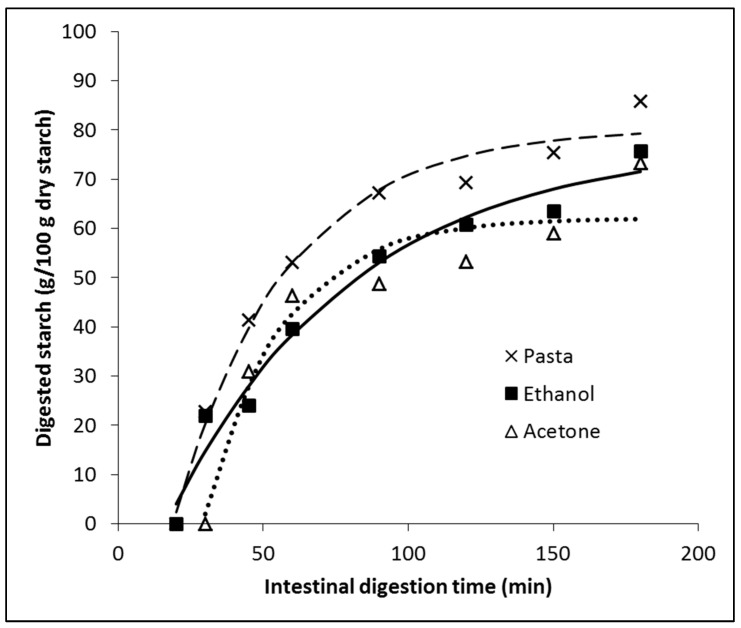
Digestograms of 5 g of pasta in the absence of extract (“Pasta”) and the presence of 20 mg of “Ethanol” or “Acetone” extracts. No digested starch was detected during the oral and gastric phases, so only the intestinal phase is shown. Points are the average of three samples. For clarity, error bars (±Standard Deviation) are not shown. The parameters of the fitted model are shown in Table 3.

**Figure 4 foods-12-03326-f004:**
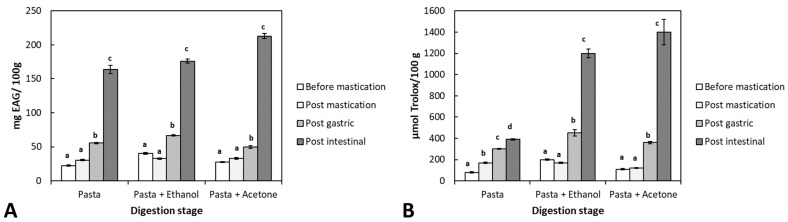
Changes of Total polyphenol content (**A**) and Antioxidant activity (**B**) on the liquid phase during in vitro digestion of pasta in the presence and absence of “Ethanol” and “Acetone” extracts. Each bar is mean ± standard deviation (*n* = 3). Different letters indicate a statistically significant difference (*p* < 0.05) for each experimental condition.

**Table 1 foods-12-03326-t001:** Kinetic parameters of α-glucosidase inhibition.

Kinetic Parameters	W/I	*D. incurvata* Extracts	
		Ethanol	Acetone
Vmax (μmol/min)	71.55	3.84 × 10^−12^	1.09 × 10^−11^
Km	0.24	4.76	5.15
Km/Vmax	0.003	1.24 × 10^12^	4.73 × 10^11^
Type of inhibition	-	Mixed	Mixed

W/I: without inhibition; Vmax: maximum velocity; Km: Michaelis–Menten constant.

**Table 2 foods-12-03326-t002:** High-resolution UHPLC-PDA-MS metabolite profiling data of the ethanolic extract of *Durvillea incurvata*.

Peak #	Retention Time (min.)	Tentative Identification	Elemental Composition [M-H]^−^	Theoretical Mass (*m*/*z*)	Measured Mass (*m*/*z*)	Accuracy (δppm)
1	2.75	Sorbitol	C_6_H_13_O_6_^−^	181.07066	181.07167	5.55
2	3.70	2,6,10,14-Tetraoxapentadecane-4,12-diol	C_28_H_45_O_11_^−^	557.27564	557.27961	3.53
3	4,20	Leucodelphinidin	C_15_H_13_O_8_^−^	321.06049	321.06188	4.30
4	5.73	Triplhoroethol	C_18_H_13_O_9_^−^	373.05685	373.05685	2.51
5	6.64	Unknown	C_13_H_27_O_8_^−^	311.17004	311.16901	−3.3
6	11.44	Hexaphloroethol or—bis-trifucophloroethol	C_36_H_25_O_8_^−^	745.10464	745.10657	4.45
7	12.03	Tetraphloroethol	C_24_H_17_O_12_^−^	497.07413	497.07364	
8	13.58	Unknown	C_14_H_29_O_8_^−^	325.18472	325.18569	−2.98
9	20.30	Hexaphloroethol or—bis-trifucophloroethol	C_36_H_25_O_8_^−^	745.10464	745.10693	5.44
10	21.75	Hexaphloroethol or—bis-trifucophloroethol	C_36_H_25_O_8_^−^	745.10464	745.10675	4.26
11	24.32	Pentaphloroethol or trifucophloroethol	C_30_H_21_O_15_^−^	621.08859	621.09015	
12	24.65	Stigmatellin	C_29_H_39_O_6_^−^	483.27412	483.27341	−1.46
13	25.25	Pentaphloroethol or trifucophloroethol	C_30_H_21_O_15_^−^	621.08750	621.08978	3.68
14	25.63	Unknown	C_28_H_51_O_11_^−^	563.34259	563.34460	3.57

**Table 3 foods-12-03326-t003:** Parameters of fitted starch digestion model, with and without the presence of seaweed extracts (during intestinal digestion; see Figure 2).

Parameters	Pasta	Ethanol	Acetone
*D*_0_ (g/100 g dry starch)	−51.14	−23.21	−123.20
*D*_∞_ (g/100 g dry starch)	80.48	77.43	62.12
*k* (1/min)	0.02606	0.01577	0.03751
*R* ^2^	0.984	0.973	0.922

## Data Availability

The data used to support the findings of this study can be made available by the corresponding author upon request.

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
