# Peer review of "Cochayuyo (*Durvillaea incurvata*) Extracts: Their Impact on Starch Breakdown and Antioxidant Activity in Pasta during In Vitro Digestion"

_foods, 2023, doi:10.3390/foods12183326_

Round 1
Reviewer 1 Report
Dear authors,
Please consider the following comments to improve the quality of manuscript:
L47-49 – Which studies? Include references and mention the recent findings.
L51 – How important is the activation? Include examples.
L63 – The process is pressurized liquid extraction (PLE). The PLE normally uses pressurized liquids coupled with heat.
L70 – Type of digestive enzyme inhibition? What do you mean?
L75 – Include city and country.
L81 – ‘’cm3’’ – cubic centimeters? Double check.
L85-L86 – Why? If authors are compared two methods of extraction, both solvents were supposed to be used.
L98 – ‘’Air bomb’’ or air pump? Double check if English writing is accurate.
L108 – use period to separate decimal places
L138 – ‘’program’’ or elution?
L143 – which column was used?
L150 ; L196; t–’’-1’’, as supercript
Topic 2.5 – how much of extract was incorporated in the pasta?
L189 – 3 as subscript
L202-L204/Eq.1 – Keep consistency in units for dimensionless
L281 – reference writing must be as [11, 14], not ‘’[11], [14]’’
L315 – same as previous
L398 – Explain how phenolics are modified in intestinal phase? Include references.
Author Response
L47-49 – Which studies? Include references and mention the recent findings.
R: Studies were included in the manuscript.
L51 – How important is the activation? Include examples.
R: It helps with glycemic homeostasis. Such idea was included in the manuscript.
L63 – The process is pressurized liquid extraction (PLE). The PLE normally uses pressurized liquids coupled with heat.
R: Yes, that is correct. In fact, the full process name is “hot pressurized liquid extraction”, such as mentioned in the manuscript.
L70 – Type of digestive enzyme inhibition? What do you mean?
R: The type of inhibition can be “competitive”, “uncompetitive” or “mixed”. We studied which one is. That means.
L75 – Include city and country.
R: They were included in the manuscript.
L81 – ‘’cm3’’ – cubic centimeters? Double check.
R: Yes, cubic centimeters.
L85-L86 – Why? If authors are compared two methods of extraction, both solvents were supposed to be used.
R: Aqueous ethanol 50% was used as eco-friendly solvent/method and was the core or the research. Acetone 60% was used as control solvent/methods. Changes in the manuscript were included for clarification.
L98 – ‘’Air bomb’’ or air pump? Double check if English writing is accurate.
R: “Air pump” is right one. It was changed in the manuscript.
L108 – use period to separate decimal places
R: It was changed in the manuscript.
L138 – ‘’program’’ or elution?
R: “Elution” is the right one. It was changed in the manuscript.
L143 – which column was used?
R: Details of the column were incorporated in the manuscript.
L150 ; L196; t–’’-1’’, as superscript
R: The text was modified properly.
Topic 2.5 – how much of extract was incorporated in the pasta?
R: It (how much and how way) is explained in the topic 2.5.1.
L189 – 3 as subscript
R: Manuscript was modified properly.
L202-L204/Eq.1 – Keep consistency in units for dimensionless
R: Consistency is kept. They are written as described by cited reference.
L281 – reference writing must be as [11, 14], not ‘’[11], [14]’’
R: References editor does not let us to do that. Journal editor probably can change that.
L315 – same as previous
R: same reply as previous.
L398 – Explain how phenolics are modified in intestinal phase? Include references.
R: Phenolics were not identified in detail during in vitro digestion in this research. However, possible modifications are discussing, and references included. For example, in the same paragraph a modification mechanism is exposed: “fragmentation of high molecular weight phlorotannins to smaller ones”.
Reviewer 2 Report
Brown seaweeds play a vital role as a food source for numerous people. Investigating the effects of their extracts on starch digestion capabilities holds significance for advancing the development of antidiabetic diets. The study focused on analyzing starch breakdown and antioxidant activity during the in vitro digestion of pasta as well as polyphenol composition in the extract was analyzed. Additionally, the composition of polyphenols in the extract was also examined.
In general, this manuscript is effectively structured and organized. It is evident that the article is commendable and exhibits only a few noteworthy concerns that warrant attention. In my view, the article has the potential for improvement by addressing the subsequent points, thereby enhancing its overall comprehensiveness.
Specific comments
To prevent confusion, it is important to maintain consistency in the usage of the terms "pasta" and "noodle."
Authors are advised to meticulously review the application of superscripts and subscripts, as inaccuracies pertaining to these elements have been identified throughout the article.
Line 26: To prevent confusion, please replace “mixed” with “mixed type”
Line 47-48: this sentence needs references.
The utilization of "ethanol" and "acetone" in figures and tables could potentially result in confusion. Consider substituting them with "ethanolic extract" and "acetonic extract," respectively, to enhance clarity.
Section 3.2: From my perspective, it is advisable to analyze the polyphenol profile in both extracts. This approach will help to address variations in the impact on starch breakdown and antioxidant activity during digestion.
I find it rather surprising that the study did not detect fucoxanthin in the extract. Could you please provide an explanation for this observation?
The quality of English in the article is generally satisfactory; however, there are a few errors present.
Author Response
To prevent confusion, it is important to maintain consistency in the usage of the terms "pasta" and "noodle."
R: When appropriate, “noodle” was changed to “pasta”. So, in general “pasta” is used throughout manuscript.
Authors are advised to meticulously review the application of superscripts and subscripts, as inaccuracies pertaining to these elements have been identified throughout the article.
R: Done.
Line 26: To prevent confusion, please replace “mixed” with “mixed type”
R: Done.
Line 47-48: this sentence needs references.
R: References were included in the manuscript.
The utilization of "ethanol" and "acetone" in figures and tables could potentially result in confusion. Consider substituting them with "ethanolic extract" and "acetonic extract," respectively, to enhance clarity.
R: In caption, "ethanol" and "acetone" mean is well defined. We consider that clarity is enough and to use more words could be inadequate.
Section 3.2: From my perspective, it is advisable to analyze the polyphenol profile in both extracts. This approach will help to address variations in the impact on starch breakdown and antioxidant activity during digestion.
R: We agree that maybe to know the polyphenol profile in both extracts would be well. However, acetonic extract was used only as control. For the objective of this research, the carryout analyses are considered adequate.
I find it rather surprising that the study did not detect fucoxanthin in the extract. Could you please provide an explanation for this observation?
R: Due to the solvents/conditions used, due to a polarity issue, it was probably not extracted.
Reviewer 3 Report
1 statistic analysis was missing in the manuscript.
2 figure 2, is this the total ion chromatogram? please specify.
3 table 2, why only three compounds having MS/MS data? For the tentative identifications, MS/MS is essential.
4 how to caculate the bioaccessibility of antioxidant compounds?
Author Response
1 statistic analysis was missing in the manuscript.
R: Topic “2.6. Statistics” was included in the manuscript.
2 figure 2, is this the total ion chromatogram? please specify.
R: Yes, it is. We think that it is clear in the manuscript.
3 table 2, why only three compounds having MS/MS data? For the tentative identifications, MS/MS is essential.
R: For clarity column was eliminated.
4 how to caculate the bioaccessibility of antioxidant compounds?
R: the antioxidant compounds in the supernatant of digestive fluid is the bioaccessible ones. For clarity, a short sentence was included in the topic 2.5.2.